# Human TRPV1 structure and inhibition by the analgesic SB-366791

Arthur Neuberger [1], Mai Oda[2], Yury A. Nikolaev[2], Kirill D. Nadezhdin [1], Elena O. Gracheva[2,3,4,5], Sviatoslav N. Bagriantsev [2] & Alexander I. Sobolevsky [1] ✉

Pain therapy has remained conceptually stagnant since the opioid crisis, which highlighted the dangers of treating pain with opioids. An alternative addiction-free strategy to conventional painkiller-based treatment is targeting receptors at the origin of the pain pathway, such as transient receptor potential (TRP) ion channels. Thus, a founding member of the vanilloid subfamily of TRP channels, TRPV1, represents one of the most sought-after pain therapy targets. The need for selective TRPV1 inhibitors extends beyond pain treatment, to other diseases associated with this channel, including psychiatric disorders. Here we report the cryo-electron microscopy structures of human TRPV1 in the apo state and in complex with the TRPV1-specific nanomolar-affinity analgesic antagonist SB-366791. SB-366791 binds to the vanilloid site and acts as an allosteric hTRPV1 inhibitor. SB-366791 binding site is supported by mutagenesis combined with electrophysiological recordings and can be further explored to design new drugs targeting TRPV1 in disease conditions.

Therapeutic control of chronic pain remains a challenging problem, with the current pain medication market dominated by agents that have been around for decades[1]. Although narcotics (opioids) are effective painkillers, they interfere with the normal brain function and cause addiction[2]. A promising alternative to opioids is the development of small molecules that target the receptors of pain, such as TRP channels[3–7]. Since temperature-induced pain is critical for survival, temperature-sensitive TRP channels (thermo-TRPs), including the founding member of the vanilloid subfamily, TRPV1, represent some of the most sought-after targets for pain therapy. Indeed, TRPV1 channels have been shown to serve as critical neuropathic pain sensors and brain inflammation detectors[8]. Altogether, brain TRPV1 channels act as detectors of harmful stimuli and key players of the microglia to neuron communication[8]. Beyond chronic pain (including nociception and neuropathic pain)[9] and thermoregulation disorders, selective high-affinity inhibitors of human TRPV1 are urgently needed in view of the channel's rapidly emerging role in various psychiatric diseases, such as depression, anxiety, fear, emotional stress, and drug abuse[10–14].

While activation by temperature, pH, the agonist capsaicin from chili peppers, and natural toxins was studied structurally in rat and squirrel TRPV1[15–19], no structures have been determined for human TRPV1, which shares only 83% sequence identity with its closest studied orthologue in rat. Similarly, structural studies of TRPV1 inhibition have been limited to low-affinity and low-specificity antagonist capsazepine applied to the rat orthologue[19]. In vitro and in vivo studies revealed high-affinity inhibition of TRPV1 by the cinnamamide antagonist SB-366791 (N-(3-methoxyphenyl)−4-chlorocinnamide), which showed little or no effect on a panel of 47 different targets, including a diverse range of G-protein-coupled receptors and voltage-gated calcium-channels[20,21]. This makes TRPV1-specific inhibitor SB-366791 a promising drug candidate for treatment of TRPV1-associated pain. Indeed, SB-366791 has been shown to effectively potentiate the analgesic effects of systemic morphine in bone cancer patients[22]. Moreover, SB-366791 has been implicated in treatment of TRPV1-governed inflammatory pain, eventually inhibiting glutamatergic transmission in a subset of neurons via a pre-synaptic TRPV1-

[1]Department of Biochemistry and Molecular Biophysics, Columbia University, New York, NY, USA. [2]Department of Cellular and Molecular Physiology, Yale University School of Medicine, New Haven, CT 06510, USA. [3]Department of Neuroscience, Yale University School of Medicine, New Haven, CT 06510, USA. [4]Program in Cellular Neuroscience, Neurodegeneration and Repair, Yale University School of Medicine, New Haven, CT 06510, USA. [5]Kavli Institute for Neuroscience, Yale University School of Medicine, New Haven, CT 06510, USA. ✉e-mail: as4005@cumc.columbia.edu

dependent mechanism following peripheral inflammation[23]. Importantly, SB-366791 was also found to impair opiate-mediated behaviors and to decrease an anxiolytic-like effect during the morphine abstinence period in rat studies, therewith offering a much-needed alternative to opioids for pain treatment without addiction side-effects[24]. A synergistic 15-fold increased antinociceptive effect was found in animal studies when SB-366791 was co-administered with a calcium channel blocker in a model of acute pain in mice[25], opening possibilities for better fine-tuning of pain therapy through combinatorial therapies.

In this study we report high-resolution single-particle cryo-electron microscopy (cryo-EM) structures of human TRPV1 in the apo state and in complex with SB-366791. We confirm the structurally identified SB-366791 binding site using whole-cell patch-clamp recordings of TRPV1-mediated currents combined with mutagenesis and propose an allosteric mechanism of TRPV1 inhibition by SB-366791. The proposed mechanism lays an important foundation for the development of much-needed new, highly specific, and potent drugs for pain therapy and treatment of other TRPV1-linked diseases.

## Results and discussion
### Functional characterization of human TRPV1
We expressed human TRPV1 (hTRPV1) in HEK 293 cells and tested its function using whole-cell patch-clamp recordings. Heating induced hTRPV1-mediated currents, which showed non-linear temperature dependence (Fig. 1a). Being nominally zero at room temperature, they showed little increase until temperature reached ~40 °C. Further heating caused dramatic increase in hTRPV1-mediated current amplitude, with the steep slope of this temperature dependence characterized by the high value of the temperature coefficient, $Q_{10} = 22.5 \pm 10.2$ (mean ± SEM, $n = 7$) (Fig. 1b). Alternatively, hTRPV1-mediated currents were induced by application of TRPV1 agonist capsaicin. In response to voltage ramps, the amplitude of the outwardly rectified hTRPV1-mediated currents became larger in the presence of increasing concentrations of capsaicin (Fig. 1c). Fitting of the current amplitude with the Hill equation gave the half maximum effective concentration of capsaicin for hTRPV1 activation, $EC_{50} = 0.234 \pm 0.059 \, \mu M$ ($n = 7$) (Fig. 1d). This value is somewhat smaller than $EC_{50}$ for rat TRPV1 ($0.527 \pm 0.053 \, \mu M$), mouse TRPV1 ($0.776 \pm 0.052 \, \mu M$), and squirrel TRPV1 ($0.529 \pm 0.032 \, \mu M$)[26], indicating that hTRPV1 is slightly more sensitive to capsaicin. Overall, hTRPV1 showed functional behavior that is consistent with previous studies of TRPV1 channels[17,26–30], and we used the same wild-type full-length construct to determine the hTRPV1 structure.

### Human TRPV1 structure in the apo state
We purified full-length hTRPV1 in lipid nanodiscs using soybean mix and synthetic lipids (POPC:POPE:POPG, 3:1:1; see Methods) and subjected these two protein samples to cryo-EM. For both samples, the micrographs showed evenly dispersed particles of hTRPV1 (Supplementary Fig. 1a, b), with 2D-class averages demonstrating clearly visible secondary structure elements, indicative of the high quality of the cryo-EM data (Supplementary Fig. 1c, d). The corresponding 3D reconstructions showed C4 symmetry (Fig. 1e) and yielded nearly identical structures, with a slightly better resolution for the sample purified in synthetic lipids (2.58 Å) compared to the soybean lipid mix (2.90 Å) (Supplementary Fig. 1e–h and Supplementary Table 1). Despite the map for the soybean lipid mix had lower resolution, it represented a more complete reconstruction of hTRPV1. Thus, we built a molecular model of hTRPV1 in soybean lipids that included residues 115–769 and excluded N-terminal residues 1–114 and C-terminal residues 770–839, which were not resolved in the cryo-EM map. At the same time, the model of hTRPV1 in synthetic lipids only included the residues 199–754. In addition, while both structures had residues 602–626 unresolved, the structure in synthetic lipids was also missing

density for residues 713–720 and the corresponding region was not included into the model. We used both structures for our analysis, but for illustrations related to the transmembrane region, we mostly used the structure obtained in synthetic lipids.

The apo-state structure of hTRPV1 (hTRPV1$_{Apo}$) has a similar fold to other representatives of the vanilloid-subfamily TRP channels. It includes a transmembrane domain (TMD) with the central ion channel pore and an intracellular skirt that is mostly built of ankyrin repeat domains (ARDs) connected by the three-stranded β-sheets and C-terminal hooks (Fig. 1f). Amphipathic TRP helices run almost parallel to the membrane and interact with the ARDs and TMDs. In addition, each ARD is connected to the TMD by a linker domain. The TMD is composed of six transmembrane helices S1–S6 and a re-entrant pore loop (P-loop) between S5 and S6. A bundle of the first four transmembrane helices represents the S1–S4 domain, which in voltage-gated ion channels forms a voltage sensor[31]. The pore domain of each subunit includes S5, P-loop and S6, and is packed against the S1-S4 domain of the neighboring subunit in a domain-swapped arrangement.

At first glance, the structure of the human ortholog of TRPV1 represented by hTRPV1$_{Apo}$ looks similar to the structures of the previously published rat and squirrel TRPV1[15,17,19,32–34] orthologs (Supplementary Fig. 2). However, their superposition gives relatively high values of the root-mean-square deviation (r.m.s.d.; 2.26 Å between human and rat orthologs and 2.39 Å between human and squirrel) because of substantial differences in the S1-S2, S2-S3, and S5-P loops as well as the C-terminus. Since these regions were reported to be involved in temperature and ligand gating of TRP channels[15,32,35–42], the observed conformational variability as well as individual substitutions in other parts of the protein may explain differences in thermosensitivity between different species and should be taken into account when designing drugs targeting TRPV1-related human diseases. For instance, a single serine substitution of the asparagine N126 in the AR1 helix of a weakly temperature-sensitive squirrel TRPV1 converts it into a highly temperature-sensitive rat TRPV1-like channel[26].

The apo-state structure of hTRPV1 shows multiple well-resolved densities of annular lipids in the TMD (Fig. 1e). Most of these densities have a clear head-and-two-tails appearance and were modeled with phosphatidylcholine (PC, Fig. 1g). An exception is the vanilloid binding site where a bulky head density justified its best fit with phosphatidylinositol (PI, Fig. 1h), consistent with the previously solved apo-state structures of rat and squirrel TRPV1[15,17,19,32–34]. Accordingly, the head of PI is accommodated by polar and charged residues of the linker domain (H411), the S2-S3 loop (D509 and S512), the S4-S5 linker (R557 and E570) and the TRP helix (Q701), while the PI tails face hydrophobic residues of S3 and S4 from one subunit, and S5 and S6 from the neighboring subunit (Supplementary Fig. 3).

### Structure of human TRPV1 in complex with inhibitor SB-366791
SB-366791 is a highly potent and specific inhibitor of TRPV1[20,21]. When co-applied with 1 μM capsaicin, 10 μM SB-366791 showed a complete inhibition of capsaicin-activated hTRPV1 (Fig. 2a, b). Current recovery from SB-366791-induced inhibition was slow and only ~50% of the current recovered after 30-s re-application of capsaicin only (Fig. 2a, c). To study the molecular mechanism of hTRPV1 inhibition by SB-366791, we solved the structure of their complex, hTRPV1$_{SB-366791}$, in GDN detergent using cryo-EM (Fig. 2d, e, Supplementary Figs. 4, 5). hTRPV1$_{SB-366791}$ has a similar overall structural architecture as hTRPV1$_{Apo}$ (Fig. 1f). Inspection of cryo-EM map revealed four identical densities with the shape of an SB-366791 molecule, one per subunit of hTRPV1 tetramer (Fig. 2f). The remarkable quality and characteristic shape of the densities allowed unambiguous modeling of the ligand position and orientation. The binding site located in the TMD region that faces the cytoplasmic leaflet of the membrane, in the crevice between S1-S4 and pore domains appears to be the vanilloid site, the

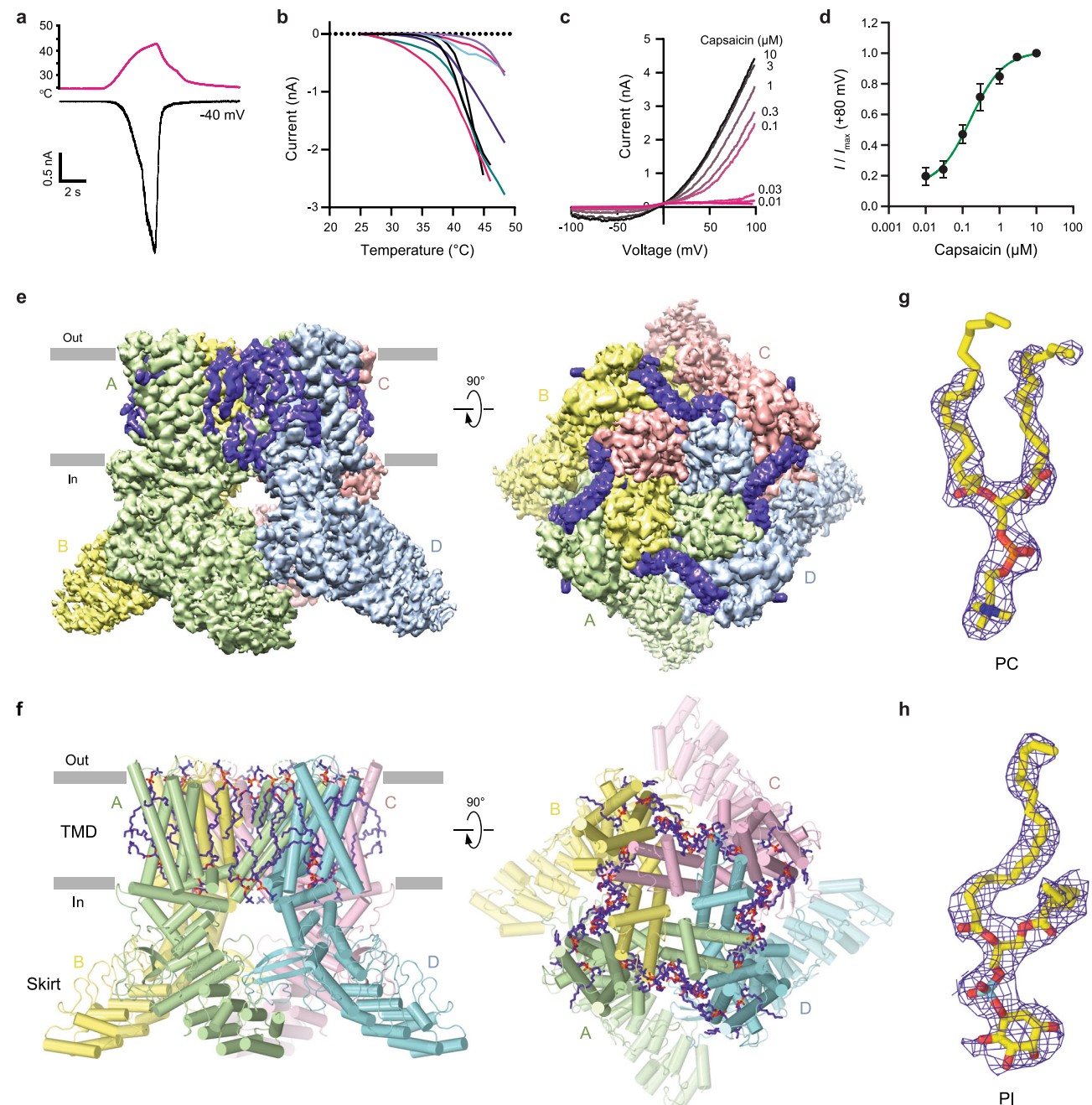

**Fig. 1 | Apo-state structure of human TRPV1. a** Representative current trace obtained at −40 mV in response to a temperature ramp from 25 °C to 45 °C recorded in the whole-cell mode from an HEK 293 cell expressing hTRPV1. **b** Current-temperature dependence obtained from recordings as in **a** showing temperature activation of hTRPV1. **c** Exemplar whole-cell current traces evoked by a voltage ramp in HEK 293 cells expressing hTRPV1 by capsaicin from 0.01 to 10 μM at 25 °C. **d** Concentration-dependence of hTRPV1 activation by capsaicin measured at +80 mV and normalized to the current at 10 μM of capsaicin. Data are shown as mean ± SEM from 7 cells and fitted with the Hill equation. Source data are provided as a Source Data file. **e** 3D cryo-EM reconstruction of hTRPV1 in the apo state viewed parallel to the membrane (left) or extracellularly (right), with density for subunits colored green, yellow, pink and blue, and lipids in purple. **f** hTRPV1$_{Apo}$ structure viewed parallel to the membrane (left) or extracellularly (right), with subunits colored similarly to **e** and lipid molecules shown in sticks. **g, h** Exemplar cryo-EM densities (purple mesh) for lipid molecules PC (**g**) and PI (**h**), shown as stick models (yellow).

same site that binds PI in the apo state as well as the agonists capsaicin, resiniferatoxin (RTX), and the competitive antagonist capsazepine (Supplementary Figs. 3 and 6)[17,19,33,34,36,41]. Binding of SB-366791 to the vanilloid site suggests that it acts as a competitive antagonist. In full agreement with the mechanism of competitive inhibition, SB-366791 caused a rightward shift in the capsaicin concentration dependence of hTRPV1-mediated calcium uptake with no apparent change in the maximal response, which was also confirmed by Schild analysis[20].

At the vanilloid site, SB-366791 forms a hydrogen bond with Y511 in S3 (Fig. 2g and Supplementary Fig. 3b). To test the importance of this interaction for SB-366791 binding, we mutated Y511 to alanine and compared the concentration dependencies of hTRPV1-mediated current inhibition for wild-type and mutant channels (Fig. 2h, i). In the rat TRPV1, the Y511A mutation was reported to be insensitive to capsaicin[43], and we observed the same for hTRPV1 (not shown). Therefore, we asked if the mutation alters sensitivity to the blocker in

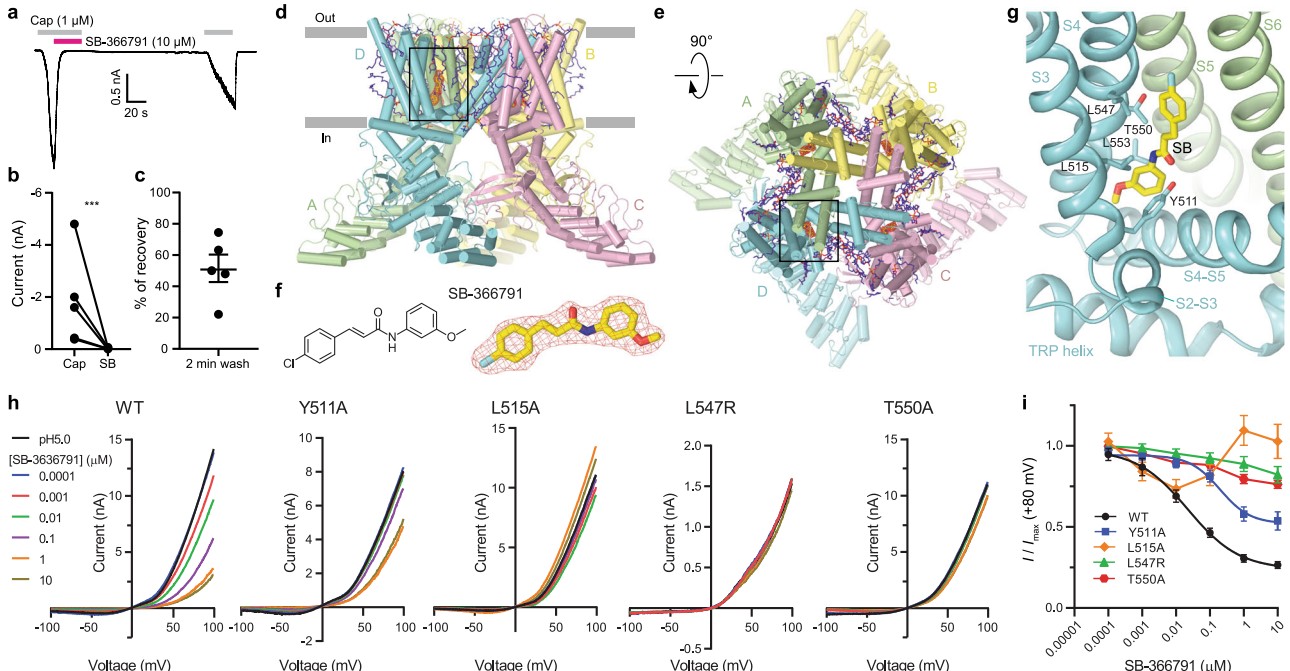

**Fig. 2 | Structure of human TRPV1 in complex with the inhibitor SB-366791.**
**a** Exemplar whole-cell current recorded at −40 mV from HEK 293 cell expressing hTRPV1 in response to application of 1 μM of capsaicin (Cap) and 10 μM of SB-366791. **b** Quantification of the inhibitory effect of 10 μM SB-366791 on hTRPV1 peak current amplitude in the presence of 1 μM capsaicin, measured 10 s after SB-366791 application. Statistical analysis: two-tailed paired $t$-test. ***$P = 0.0003$. Data points represent paired recordings from 5 individual cells. **c** Quantification of capsaicin-stimulated hTRPV1 peak current amplitude recovery after termination of SB-366791 application. The cell was washed for at least 2 min before the second application of Cap. Data are shown as mean ± SEM. Data points represent recordings from 5 individual cells. **d**, **e** hTRPV1$_{SB-366791}$ structure viewed parallel to the membrane (**d**) or extracellularly (**e**), with subunits colored green, yellow, pink, and

blue, and lipids shown as purple sticks. **f** Chemical structure (left) and molecular model (right) of SB-366791, with the cryo-EM density for the inhibitor shown as red mesh. **g** Close-up view of the binding pocket, with SB-366791 and residues that contribute to its binding shown in sticks. **h** Exemplar whole-cell currents recorded from hTRPV1-expressing HEK 293 cell in response to a voltage ramp in the presence of different concentrations of SB-366791 at pH 5. **i** Concentration-dependencies of SB-366791 inhibition of wild-type and mutant hTRPV1 channels at pH 5, measured at +80 mV and normalized to the value of pH5-induced current. Data are shown as mean ± SEM. Lines represent fits to the Hill equation (WT, $n = 7$; Y511A, $n = 7$) or connect data points (L515A, $n = 7$; L547R, $n = 7$, T550A, $n = 7$). Source data are provided as a Source Data file.

response to lowering pH, a known TRPV1 activating stimulus. We found that, compared to wild-type, the Y511A mutation lowered sensitivity to SB-366791 by augmenting the $IC_{50}$ value from $0.021 ± 0.006$ μM ($n = 7$) for the wild-type to $0.253 ± 0.075$ μM ($n = 7$, $P = 0.0097$, $t$-test) for the mutant, and increased the proportion of blocker-resistant current from $0.220 ± 0.027$ ($n = 7$) to $0.490 ± 0.070$ ($n = 7$, $P = 0.0034$, $t$-test) (Fig. 2h, i). SB-366791 also forms hydrophobic interactions with L515 in S3 and L547, T550 and L553 in S4 (Fig. 2g). Strikingly, disrupting the hydrophobic interactions by L515A, L547R and T550A mutations rendered the channel completely insensitive to inhibition by the blocker (Fig. 2h, i). Together with the structural data, these results strongly support the idea that hTRPV1 inhibition by SB-366791 is due to binding of this ligand to the vanilloid site.

## Closed pore of human TRPV1 structures

The pore of hTRPV1 in the apo and SB-366791-bound structures has two narrow constrictions, one in the selectivity filter formed by the backbone carbonyl oxygen of G644 and another one in the gate region of S6 bundle crossing formed by the side chains of I680 (Fig. 3a). Measurements of the pore radius suggest that the pore has approximately the same size in all three hTRPV1 structures and the constriction formed by the I680 side chains hydrophobically seals the pore and prevents ion conductance (Fig. 3b). Previously, a similar extent of pore closure in the gate region was reported in the closed-state structures of rat and squirrel TRPV1 solved in the apo conditions or in the presence of inhibitors[15,17,19,32–34]. Compared to the closed-pore conformations of hTRPV1$_{Apo}$ and hTRPV1$_{SB-366791}$, the pore of the open-state rTRPV1

undergoes widening in both the selectivity filter and gate regions, sufficient to allow water and ion conductance[32,34] (Fig. 3b). Because SB-366791 inhibits hTRPV1-mediated current in the presence of capsaicin (Fig. 2a), its competitive character assumes the replacement of capsaicin with SB-366791 in the vanilloid binding pocket. How does the antagonist molecule of about the same size as the agonist molecule cause hTRPV1 channel closure upon binding to the vanilloid site?

## Structural changes accompanying competitive inhibition of hTRPV1 by SB-366791

To highlight structural changes and regions of TRPV1 involved in competitive inhibition by SB-366791, we made a detailed comparison of hTRPV1$_{SB-366791}$ with one of the recently published open-state structures of rTRPV1 activated by RTX[32] (Fig. 4). Calculation of the r.m.s.d. values for the $C_α$ atoms with a short sliding window of 10 residues highlighted the structural regions that undergo drastic secondary structure rearrangements (Fig. 4a, b). Expectedly, strong changes involved S4 to the S4-S5 linker connection as well as the S2-S3 loop, the regions contributing to the vanilloid site, the site of binding of both the inhibitor SB-366791 and the agonist RTX. Exchange of the ligands between the open and inhibited states resulted in a slight bending of S5 and, as a result, of the adjacent S6, highlighted by coloring of the bundle crossing of these helical segments (Fig. 4a, b). Most significantly, the inhibition-associated changes in the secondary structure are culminated in the region connecting S6 and the TRP helix. These strongest changes are caused by SB-366791 inhibition-induced shortening of S6 and elongation of the TRP helix.

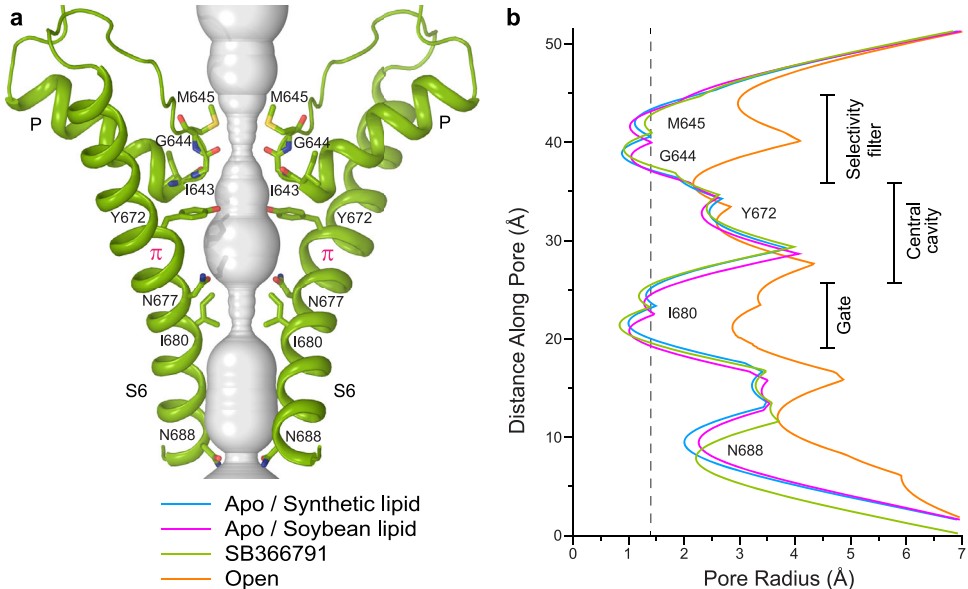

**Fig. 3 | Closed pore of hTRPV1$_{Apo}$ and hTRPV1$_{SB-366791}$ structures. a** Pore-forming domain in hTRPV1$_{SB-366791}$ with the residues contributing to pore lining shown as sticks. Only two of four subunits are shown, with the front and back subunits omitted for clarity. The pore profile is shown as a space-filling model (gray). The region that undergoes α-to-π transition in S6 is labeled (π). **b** Pore radius for hTRPV1$_{Apo}$ in synthetic (blue) and soybean (pink) lipids, hTRPV1$_{SB-366791}$ (green) and rTRPV1$_{Open}$ (orange; PDB ID: 7RQW) calculated using HOLE. The vertical dashed line denotes the radius of a water molecule, 1.4 Å.

As a measure of structural movement associated with the change in TRPV1 conformational state, we calculated $C_\alpha$ atom translations between the corresponding structures superposed based on the TMDs (Fig. 4c, d). Interestingly, while the ligand binding region shows minimal overall structural movements, the agonist-to-inhibitor exchange results in a dramatic movement of S6 and the TRP helix – the major structural regions involved in TRPV1 gating. Surprisingly, substantial translation was also observed for S1, the peripheral TMD segment, which adapts a slightly different position between human and rat TRPV1 and is unlikely to play a major role in SB-366791-induced inhibition. As we localized the structural changes associated with TRPV1 competitive inhibition by SB-366791, we looked at detailed molecular interactions that underlie these structural changes and are integral to the molecular mechanism of TRPV1 inhibition.

## Putative mechanism of TRPV1 competitive inhibition by SB-366791

When binding to the vanilloid site, SB-366791 does not reach as deep intracellularly as the agonist RTX (Fig. 5a). When RTX reaches deep into the vanilloid site, its hydroxyl group interacts with R557 in S4 by pulling it up extracellularly. In this new position, R557 is 3.2 Å away from E570 in S5 and capable of forming a salt bridge with this glutamate – the interaction proposed to be crucial for TRPV1 activation[27]. The R557 – E570 salt bridge stabilizes a closer position of S5 to the S1-S4 domain, which is further enforced by the interaction of R579 and Q561 from the S4-S5 segments of the neighboring subunits, in turn guided by the flipping side chain of Y565 in response to tightening of the S4-S5 interface. On the other hand, when SB-366791 is bound to the vanilloid site, it does not interact with R557, and this arginine remains unengaged into the interaction with E570 (the closest distance between them 6.1 Å). Correspondingly, R579 and Q561 remain too distal from each other to be able to interact. Instead, the SB-366791-bound as well as the closed apo states are likely to be stabilized by a salt bridge between R575 in S5 and E693 in the TRP helix (the distance between them <4 Å), which is missing in the open state (where the distance >12 Å) (Fig. 5a). These two sets of molecular interactions are likely to promote positioning of S6 more peripheral or proximal to the channel axis and result in opening (Fig. 5b) or closure (Fig. 5c) of

the lower gate, respectively. Another competitive antagonist of TRPV1, capsazepine, which also binds to the vanilloid site[19] (Supplementary Fig. 3), most probably acts similarly to SB-366791 since the R557-E570 distance in the capsazepine-bound rat TRPV1 (~4.6 Å; PDB ID: 5IS0) is too long, while the R575-E692 distance (<4 Å) is sufficiently short to allow the corresponding electrostatic interactions.

A comparison of the closed apo, open and SB-366791-bound inhibited states outlines the putative mechanisms of TRPV1 activation and competitive inhibition (Fig. 5d). When the agonist RTX binds to the vanilloid site (Closed to Open transition in Fig. 5d), it engages R557 into a salt bridge with E570. This interaction pulls S5 towards S1-S4 domain. The tightening of the S4-S5 interface flips the side chain of Y565 and promotes the hydrogen bond between R579 and Q561. These two interactions stabilize a remote position of S5 relative to the channel pore, which eliminates the closed-state stabilizing interaction between R575 and E693 and pulls the adjacent S6 segment away from the channel pore as well. As a result of S6 movement, the lower gate at the S6 bundle crossing opens and allows water and cations to permeate the pore.

When the competitive inhibitor SB-366791 replaces the agonist at the vanilloid site (Open to Inhibited transition in Fig. 5d), its methoxy group positions at the bottom of the binding pocket. This methoxy group, however, lacks the ability to form a hydrogen bond with R557. Correspondingly, SB-366791 likely pushes away the side chain of R557 and disrupts its salt bridge with E570. S5 slips away from S4 accompanied by disruption of the hydrogen bond between R579 and Q561. The increased space between S4 and S5 allows Y565 to flip its side chain and fill up the interface. At the same time, R575 and E693 become close enough to form a salt bridge, which further stabilizes the closer to the channel pore position of S5, and S6 correspondingly. Such change in S6 position tightens the S6 bundle and closes the lower gate for conduction of ions and water. To test the importance of the proposed interaction between R575 and E693 for the hTRPV1 inhibition by SB-366791, we introduced an alanine substitution of E693 (Supplementary Fig. 7a–c). While activation by capsaicin remained virtually unchanged ($EC_{50} = 0.207 \pm 0.029$ μM, $n = 6$ for E693A compared to $EC_{50} = 0.234 \pm 0.059$ μM, $n = 7$ for WT, $P = 0.709$, t-test), inhibition by SB-366791 was significantly weakened by the mutation due to reduced

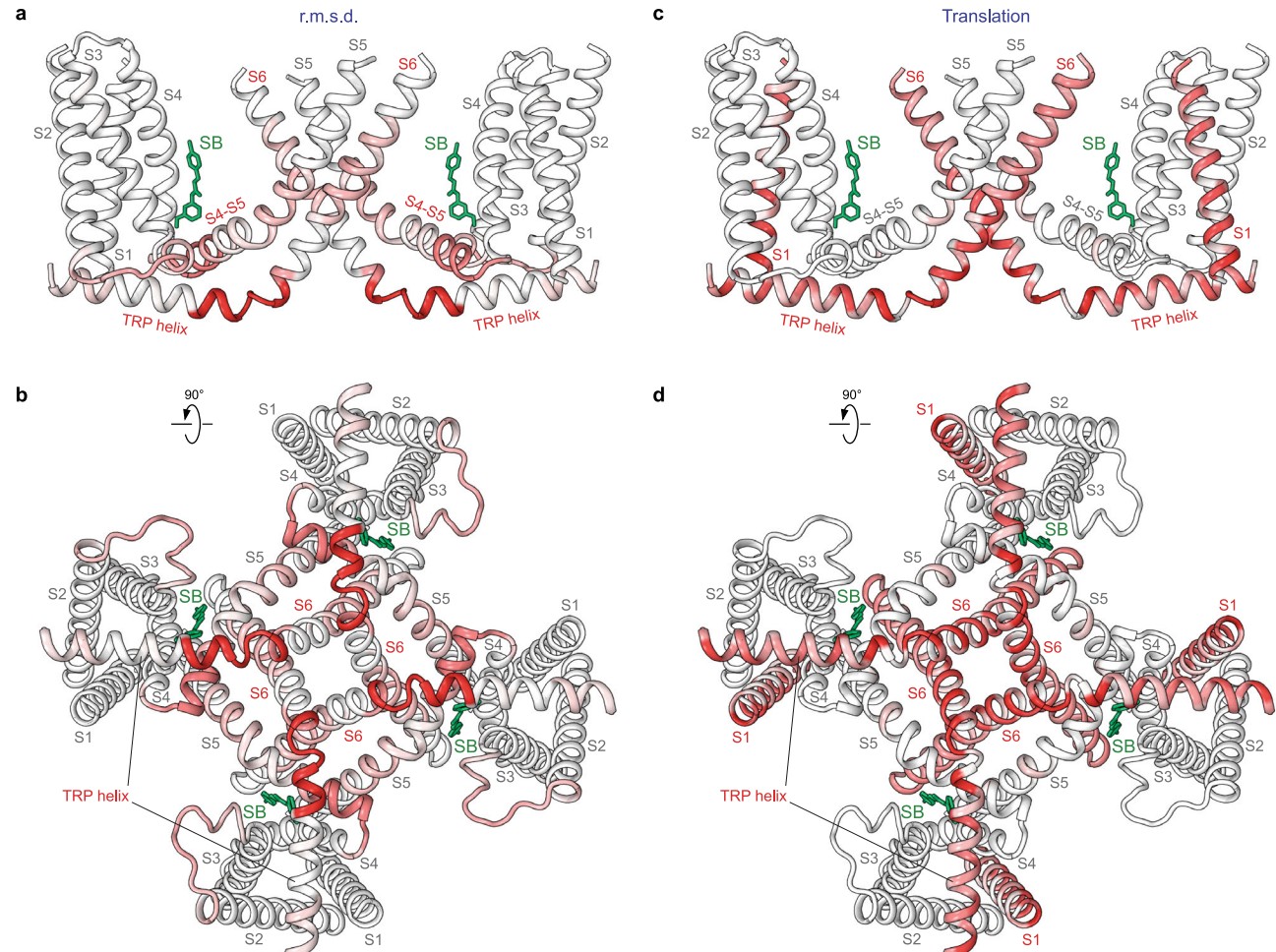

**Fig. 4 | Structural changes accompanying competitive inhibition of hTRPV1 by SB-366791. a–d** Gray (no changes) to red (strong changes) gradient of the r.m.s.d. (**a**, **b**) or translation (**c**, **d**) calculated between the inhibited (hTRPV1$_{SB-366791}$) and open (rTRPV1, PDB ID: 7RQW) states and mapped on the hTRPV1$_{SB-366791}$ TMD. Only two of four subunits are shown in **a** and **c**, with the front and back subunits omitted for clarity. Molecules of SB-366791 are shown in sticks (green). The r.m.s.d. values were calculated for C$_\alpha$ atoms along the entire TRPV1 sequence with a sliding window of 10 residues. C$_\alpha$ atom translations were calculated after aligning the TMDs of the corresponding structures. Regions of the greatest structural changes are labeled in red.

potency ($IC_{50} = 0.100 \pm 0.011\,\mu M$, $n = 6$ for E693A compared to $IC_{50} = 0.021 \pm 0.006\,\mu M$, $n = 7$ for WT, $P < 0.0001$, $t$-test) and the elevated fraction of blocker-insensitive current (E693A, $0.440 \pm 0.034$, $n = 6$; WT, $0.22 \pm 0.03$, $n = 7$; $P = 0.0004$, $t$-test, Supplementary Fig. 7d–g), strongly supporting the proposed structural mechanism (Fig. 5).

Competitive inhibition by SB-366791 (Fig. 5) does not necessarily require TRPV1 activation and ion channel opening. Even in the absence of an agonist (capsaicin or RTX), when the channel is in the closed apo state, SB-366791 can possibly bind to the vanilloid site by out-competing the phosphatidylinositol (PI) lipid, thus stabilizing the closed conformation. On the other hand, the agonist-bound channel is also likely to dynamically visit TRPV1 conformations in which the channel is either open or closed. SB-366791 may replace the agonist while the channel is in either conducting or non-conducting state. The state-dependence of the SB-366791 efficiency to bind to the vanilloid site remains an open question and answering it will require further experimentation.

Pharmacological and physiological investigations of TRPV1 have so far been focused on rat and squirrel orthologues whilst no structural and almost no pharmacological or physiological data was collected for human TRPV1 despite that (1) human TRPV1 shares only 83% sequence identity with its closest studied orthologue in rat and (2) the channel plays a central role in almost every aspect of human physiology and disease including pain and temperature perception. TRPV1 channels are critical neuropathic pain sensors and brain inflammation detectors[8]. The revealed mechanism of TRPV1 inhibition by a TRPV1-specific drug in development allows us to understand the delicate allosteric modulation of TRPV1 on near-atomic level. These findings open new avenues for the design of targeted and selective next-generation analgesic drugs as well as drugs targeting other TRPV1-related diseases.

## Methods
### Expression and purification
Human TRPV1 was expressed and purified as previously described in detail for TRPV3[44–46] with slight modifications. In short, bacmids and baculoviruses were produced using a standard method[47]. Baculovirus was made in Sf9 cells (Thermo Fisher Scientific, mycoplasma test negative, GIBCO #12659017) for ~72 h and was added to suspension-adapted HEK 293 S cells lacking N-acetyl-glucosaminyltransferase I (GnTI⁻, mycoplasma test negative, ATCC #CRL-3022) that were maintained at 37 °C and 5% $CO_2$ in Freestyle 293 media (Gibco-Life Technologies #12338-018) supplemented with 2% FBS. To enhance protein expression, sodium butyrate (10 mM) was added 12 h after transduction and the temperature was reduced to 30 °C. At 48–72 h post-transduction, the cells were harvested by centrifugation at 5,471 g for 15 min using a Sorvall Evolution RC centrifuge (Thermo Fisher

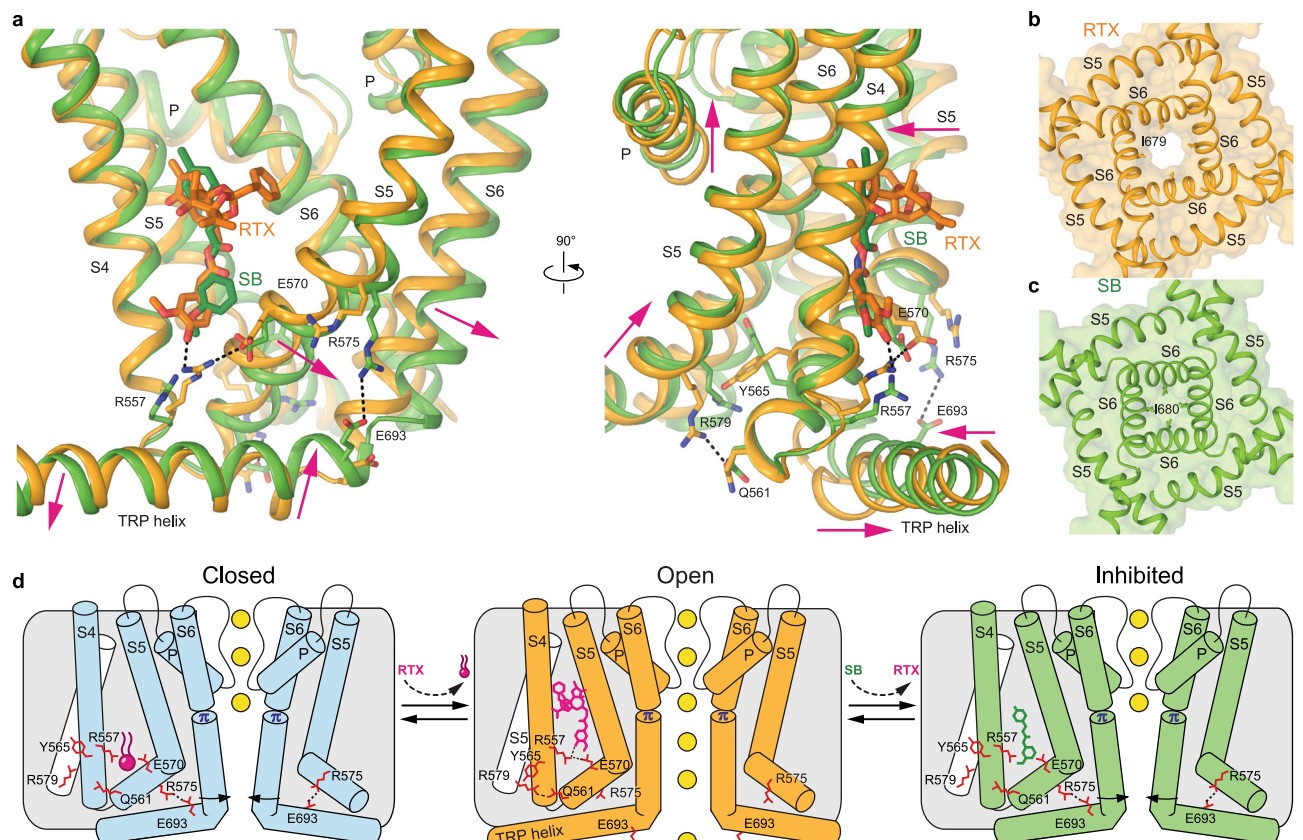

**Fig. 5 | Mechanism of hTRPV1 inhibition by SB-366791. a** Closeup view of the vanilloid site in superposed RTX-bound open rTRPV1 (orange; PDB ID: 7RQW) and inhibited hTRPV1$_{SB-366791}$ (green) structures. Relative movements of domains are indicated by pink arrows. **b**, **c** Intracellular view of the gate region in RTX-bound open rTRPV1 (**b**) and inhibited hTRPV1$_{SB-366791}$ (**c**) structures. **d** Schematic mechanism of TRPV1 activation and inhibition by SB-366791.

Scientific), washed in phosphate buffer saline (PBS) pH 8.0, and pelleted by centrifugation at 3,202 g for 10 min using an Eppendorf 5810 centrifuge. The cell pellet was resuspended in the ice-cold buffer containing 20 mM Tris pH 8.0, 150 mM NaCl, 0.8 µM aprotinin, 4.3 µM leupeptin, 2 µM pepstatin A, 1 µM phenylmethylsulfonyl fluoride (PMSF), and 1 mM β-mercaptoethanol (βME). The suspension was then supplemented with 2% (w/v) digitonin and cells were lysed at constant stirring for 2 h at 4 °C. Unbroken cells and cell debris were pelleted in an Eppendorf 5810 centrifuge at 3,202 g and 4 °C for 10 min. Insoluble material was removed by ultracentrifugation for 1 h at 186,000×*g* in a Beckman Coulter centrifuge using a Ti-45 type rotor. The supernatant was added to the strep resin, which was then rotated for 1 h at 4 °C. The resin was washed with 10 column volumes of wash buffer containing 20 mM Tris pH 8.0, 150 mM NaCl, 1 mM βME, and 0.01% (w/v) glyco-diosgenin (GDN), and the protein was eluted with the same buffer supplemented with 2.5 mM D-desthiobiotin. The eluted protein was concentrated to 0.5 ml using a 100-kDa NMWL centrifugal filter (MilliporeSigma™ Amicon™) and then centrifuged in a Sorvall MTX 150 Micro-Ultracentrifuge (Thermo Fisher Scientific) using a S100AT4 rotor for 30 min at 66,000 ×*g* and 4 °C before being injected into a size-exclusion chromatography (SEC) column. The protein was purified using a Superose™ 6 10/300 GL SEC column attached to an AKTA FPLC (GE Healthcare) and equilibrated with the buffer containing 150 mM NaCl, 20 mM Tris pH 8.0, 1 mM βME, and 0.01% (w/v) GDN. The tetrameric peak fractions were pooled and concentrated using a 100-kDa NMWL centrifugal filter (MilliporeSigma™ Amicon™) to 3.14 mg/ml. SB-366791 was added to hTRPV1 at the concentration of 121 µM.

hTRPV1 in apo state was reconstituted in circularized NW11 nanodiscs (cNW11). cNW11 were prepared according to the standard

protocol[37,48,49] and stored at –80 °C as ~2–3-mg/ml aliquots in the buffer containing 20 mM Tris pH 8.0 and 150 mM NaCl before usage. For nanodisc reconstitution, the purified protein was mixed with cNW11 nanodiscs and either POPC:POPE:POPG (3:1:1; Avanti Polar Lipids) or soybean lipids (Soy polar extract, Avanti Polar Lipids) at the molar ratio of 1:3:166 (TRPV1:cNW11:lipid). The lipids were dissolved in the buffer containing 150 mM NaCl and 20 mM Tris pH 8.0 to reach the concentration of 100 mg/ml and subjected to 5–10 cycles of freezing in liquid nitrogen and thawing in a water bath sonicator. The nanodisc mixture (500 µl) was rocked at room temperature for 1 h. Subsequently, 40 mg of Bio-beads SM2 (Bio-Rad), pre-wet in the buffer containing 20 mM Tris pH 8.0 and 150 mM NaCl, were added to the nanodisc mixture, which was then rotated for one hour at 4 °C. After adding 40 mg more of Bio-beads SM2, the resulting mixture was rotated at 4 °C for another ~14 h. The Bio-beads SM2 were then removed by pipetting. The sample was then centrifuged in a Sorvall MTX 150 Micro-Ultracentrifuge (Thermo Fisher Scientific) using a S100AT4 rotor for 30 min at 66,000 × g and 4 °C before being injected into a SEC column. Nanodisc-reconstituted hTRPV1 was then purified from empty nanodiscs using SEC with a Superose™ 6 10/300 GL SEC column equilibrated with the buffer containing 150 mM NaCl, 20 mM Tris pH 8.0, and 1 mM βME. Fractions of nanodisc-reconstituted hTRPV1 were pooled and concentrated to 2.5 mg/ml using a 100-kDa NMWL centrifugal filter. 0.0005% LMNG was added to the specimen for better distribution of particle orientations.

**Cryo-EM sample preparation and data collection**

UltrAuFoil 1.2/1.3, Au-50 (300-mesh) grids were used for plunge-freezing. Prior to sample application, grids were plasma treated in a PELCO easiGlow glow discharge cleaning system (0.39 mBar, 15 mA,

"glow" for 25 s, and "hold" for 10 s). A Mark IV Vitrobot (Thermo Fisher Scientific) set to 100% humidity at 4 °C was used to plunge-freeze the grids in liquid ethane after applying 3 μl of protein sample to their gold-coated side using a blot time of 5 s, a blot force of 5, and a wait time of 15 s. The grids were stored in liquid nitrogen before imaging.

Images of frozen-hydrated particles of hTRPV1$_{SB-366791\_GDN}$ were collected on a Titan Krios transmission electron microscope (Thermo Fisher Scientific) operating at 300 kV and equipped with a post-column GIF Quantum energy filter and a Gatan K3 Summit direct electron detection camera (Gatan, Pleasanton, CA, USA) using EPU software (Thermo Fisher Scientific). 4188 micrographs were collected in super-resolution mode with raw image pixel size of 0.34 Å across the defocus range of −0.75 to −1.5 μm. The total dose of ~60 e⁻Å⁻² was attained by using the dose rate of ~15 e⁻pixel⁻¹s⁻¹ across 50 frames during the 1.87-s exposure time.

Images of frozen-hydrated particles of hTRPV1$_{Apo\_cNW11\_POPC:POPE:POPG}$ were collected on a Titan Krios transmission electron microscope (Thermo Fisher Scientific) operating at 300 kV and equipped with a post-column GIF Quantum energy filter and a Gatan K3 Summit direct electron detection camera (Gatan, Pleasanton, CA, USA) using SerialEM[50]. 16,062 micrographs were collected in super-resolution mode with raw image pixel size of 0.4 Å across the defocus range of −0.8 to −2.0 μm. The total dose of ~60 e⁻Å⁻² was attained by using the dose rate of ~16 e⁻pixel⁻¹s⁻¹ across 50 frames during the ~2-s exposure time.

Images of frozen-hydrated particles of hTRPV1$_{Apo\_cNW11\_Soybean}$ were collected on a Titan Krios transmission electron microscope (Thermo Fisher Scientific) operating at 300 kV and equipped with a post-column GIF Quantum energy filter and a Gatan K3 Summit direct electron detection camera (Gatan, Pleasanton, CA, USA) using EPU software. 19,303 micrographs were collected in counting mode with raw image pixel size of 0.873 Å across the defocus range of −0.75 to −2.0 μm. The total dose of ~50 e⁻Å⁻² was attained by using the dose rate of ~13.6 e⁻pixel⁻¹s⁻¹ across 40 frames during the 2.8-s exposure time.

### Image processing and 3D reconstruction

Data were processed in cryoSPARC[51] and RELION[52]. Movie frames were aligned using the patch motion correction in cryoSPARC or MotionCor2[53] implemented in RELION. Contrast transfer function (CTF) estimation was performed on non-dose-weighted micrographs using the patch CTF estimation. Subsequent data processing was done on dose-weighted micrographs. Following CTF estimation, micrographs were manually inspected and those with outliers in defocus values, ice thickness, and astigmatism as well as micrographs with lower predicted CTF-correlated resolution (higher than 5 Å) were excluded from further processing (individually assessed for each parameter relative to the overall distribution). After several rounds of selection through 2D classification, particles were further 3D classified (heterogeneous refinement) into four classes. Particles representing the best class were re-extracted without binning (256-pixel box size) and further 3D classified. The final set of 67,470, 778,428, and 323,292 particles for hTRPV1$_{SB-366791\_GDN}$, hTRPV1$_{Apo\_cNW11\_POPC:POPE:POPG}$, and hTRPV1$_{Apo\_cNW11\_Soybean}$, respectively, representing the best class were subjected to a series of refinements including homogenous, non-uniform, and CTF refinements. The reported resolution of 2.29 Å, 2.58 Å, and 2.90 Å for the final maps of hTRPV1$_{SB-366791\_GDN}$, hTRPV1$_{Apo\_cNW11\_POPC:POPE:POPG}$, and hTRPV1$_{Apo\_cNW11\_Soybean}$, respectively, were estimated using the gold standard Fourier Shell Correlation (FSC). The local resolution was calculated with the resolution range estimated using the FSC = 0.143 criterion. Cryo-EM density visualization was done in UCSF Chimera[54] and UCSF ChimeraX[55].

### Model building

To build models of hTRPV1 in Coot[56], we used the previously published cryo-EM structures of TRPV1[17] as guides. The models were tested for overfitting by shifting their coordinates by 0.5 Å (using Shake) in Phenix[57], refining each shaken model against the corresponding unfiltered half map, and generating densities from the resulting models in UCSF Chimera. Structures were visualized and figures were prepared in UCSF Chimera, UCSF ChimeraX[55], and Pymol[58]. The pore radius was calculated using HOLE[59].

### Electrophysiology

100 ng of hTRPV1 constructs (wild type, Y511A, L515A, L547R, T550A, and E693A) cloned into pIRES-EGFP[60] were co-transfected with 900 ng of pcDNA3.1(-) plasmid into HEK293T$^{\Delta PIEZO1}$ cells using the Lipofectamine 3000 reagent (Thermo Fisher). 12–16 h after transfection, cells were plated onto coverslips coated with Matrigel Matrix (BD Bioscience) and analyzed by voltage-clamp electrophysiology 2 h after plating. Whole-cell recordings were performed in the bath solution containing (in mM): 140 NaCl, 5 KCl, 10 HEPES, 2 MgCl$_2$, 2 CaCl$_2$, 10 glucose, pH 7.4 (pH adjusted with NaOH), with 1–3 MΩ resistance electrodes filled with the pipette solution containing (in mM): 150 KCl, 3 MgCl$_2$, 10 HEPES, 5 EGTA, pH 7.2 (pH adjusted with KOH), using the Axopatch 200B amplifier and pClamp 10.3 software suite (Molecular Devices). Currents were evoked by voltage ramp from −100 mV to +100 mV every 1 s, from a holding potential of −40 mV, filtered at 1 kHz and sampled at 2 kHz using the Digidata 1440 A digitizer (Molecular Devices). The pH 5 solution was the bath solution made with MES instead of HEPES. SB-366791 (Cayman Chemical Company) was dissolved in DMSO for a stock concentration of 18 mM, stored at −20 °C and diluted to the working concentration prior to experiment. To plot concentration-dependence curves, whole-cell currents were measured at +80 mV, data from each cell were normalized by the value of agonist-induced current, combined and fitted to the modified Hill equation. Data from individual fits were combined to perform statistical analysis of $EC_{50}$, $IC_{50}$ and minimal current values.

### Reporting summary

Further information on research design is available in the Nature Portfolio Reporting Summary linked to this article.

## Data availability

All data needed to evaluate the conclusions of the paper are present in the paper or the Supplementary Information. The cryo-EM density maps of hTRPV1 in the apo state and in complex with SB-366791 were deposited to the Electron Microscopy Data Bank (EMDB) under the accession codes EMD-29981 (hTRPV1$_{Apo}$; cNW11, soybean lipids), EMD-29982 (hTRPV1$_{Apo}$; cNW11, POPC:POPE:POPG), and EMD-29983 (hTRPV1$_{SB-366791}$), respectively. The atomic coordinates have been deposited to the Protein Data Bank (PDB) under the accession codes 8GF8 (hTRPV1$_{Apo}$; cNW11/soybean lipids), 8GF9 (hTRPV1$_{Apo}$; cNW11/POPC:POPE:POPG), and 8GFA (hTRPV1$_{SB-366791}$) (see Supplementary Table 1), respectively. All other data are available from the corresponding author upon request. Source data are provided with this paper.

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

## Acknowledgements

We thank Guobin Hu and Jake Kaminsky (Laboratory for BioMolecular Structure), Sean Mulligan (Pacific Northwest Center for Cryo-EM) as well as Thomas Edwards, Adam Wier, and Tara Fox (National Cancer Institute, Frederick National Laboratory) for help with microscope operation and data collection. Some of this work was performed at the Laboratory for BioMolecular Structure (LBMS) and supported by the DOE Office of Biological and Environmental Research (KP1607011). A portion of this research was supported by NIH grant U24GM129547 and performed at the PNCC at OHSU and accessed through EMSL (grid.436923.9), a DOE Office of Science User Facility sponsored by the Office of Biological and Environmental Research. This research was, in part, supported by the National Cancer Institute's National Cryo-EM Facility at the Frederick National Laboratory for Cancer Research under contract HSSN261200800001E. A.N. is a Walter Benjamin Fellow funded by the Deutsche Forschungsgemeinschaft (DFG, German Research Foundation) – 464295817. S.N.B. was supported by grants from NIH (R01NS097547, R01NS126277) and NSF (1923127, 2114084). A.I.S. was supported by the NIH (R01 AR078814, R01 CA206573, R01 NS083660, R01 NS107253) and NSF (1818086).

## Author contributions

A.N. carried out protein expression, protein purification, and cryo-EM data processing. M.O. and Y.A.N. carried out electrophysiological experiments. A.N. and K.D.N. made constructs and prepared cryo-EM samples. A.I.S. built atomic models. A.N., M.O., Y.A.N., K.D.N., E.O.G., S.N.B., and A.I.S. contributed to the preparation of the manuscript.

## Competing interests

The authors declare no competing interests.
