## [Peer Review File · Nature Communications]

Human TRPV1 structure and inhibition by the analgesic
SB-366791REVIEWER COMMENTS

Reviewer #1 (Remarks to the Author):

This article by Neuberger et al. reports three cryoEM structures of the human TRPV1 channel for the first time in apo and inhibitor-bound states. The structures reveal the SB-366791 binding pocket, which is validated by mutagenesis and electrophysiology. Based on these observations and structural analyses, a mechanism of SB-366791 inhibition is proposed. The experiments are carefully done and the manuscript is well written. Although several structures of TRPV1 homologs have already been published, this study provides very exciting new structural insights into the structural pharmacology of human TRPV1. Given the importance of TRPV1 in human biology and its emerging role in pain modulation, I recommend the publication of this work.

I only have a few minor comments for the authors to consider adding into the discussion.

1. How similar is the apo state of human TRPV1 compared to that of other homologs in terms of RMSD?
2. Line 166. What is the rationale to mutate L547 into Arg but not Ala (while mutating Y511 and L515 into Ala)?
3. Is the proposed inhibition mechanism of SB-366791 different from that of capsaizepine?
4. It is definitely not necessary to re-collect a dataset, but is there a reason why the SB-366761-bound structure is determined in detergent, rather than in nanodisc?

Chia-Hsueh Lee

Reviewer #2 (Remarks to the Author):

The TRPV1 is a cation-selective ion channel that is expressed in pain-sensing neurons and is activated by heat, vanilloids, protons, and by inflammatory conditions. TRPV1 inhibitors could potentially be used as non-addictive analgesics by stopping some forms of pain at the source – the nociceptive neurons. Multiple cryo-EM structures of TRPV1 of a few animal species have been obtained in the presence of an antagonist and agonists that bind to the vanilloid pocket. In the present manuscript, a structure of the full-length human TRPV1 channel is obtained in the apo state and in the presence of the antagonist SB-

366791. The structures are of high quality, displaying well-defined maps for multiple lipids that had been previously observed. There is also a clear non-protein density at the vanilloid pocket that can be assigned to SB-366791, consistent with previous studies showing that SB-366791 acts as a competitive inhibitor for activation by the vanilloid agonist capsaicin. Although the structural work is solid, and the data of high quality, the new insight provided by this manuscript is limited, and in my opinion would not warrant publication in its present form. My specific concerns are as follows:

1. I recognize the potential value of having a structure of the human TRPV1 channel, as opposed to structures of rodent orthologues. However, the human and rat TRPV1 channels are almost identical (83% sequence identity). Importantly, the regions that contain the highest number of amino acid sequence differences between human and rodents seem to be located at regions that were not well defined in the structures in this manuscript. No discussion is provided to address whether there are indeed any significant differences between the human structures and those of animals, and if any of these differences could be of any mechanistic or biological relevance. If no noteworthy differences can be found in the regions that were well-defined in maps, I would argue that the advancement represented by these human structures is rather incremental.

2. There was already evidence pointing to SB-366791 binding to the vanilloid pocket, and also showing that it also antagonizes channel activation by acidic pH. The mutagenesis results in Figure 2 are therefore unsurprising, especially considering that those same mutations also affect activation by vanilloids as shown in this and other studies. The specific binding pose identified in the experimental map in this study could be valuable. However, the conformation of the channel bound to SB-366791 appears to be indistinguishable from the apo state of the human and rodent structures, as well as the structure bound to the inhibitor capsazepine that also targets the vanilloid site – no discussion is provided to indicate the extent to which these structures are similar to each other, but from the cartoon in Figure 5d, the same interactions are highlighted in apo and SB-366791-bound states. Therefore, the specific mechanism of inhibition of SB-366791 remains in question: whereas it could act competitively to displace vanilloid agonists, it is less clear how it would inhibit channel activation by protons and other non-vanilloid activators. Structures of TRPV1 have been obtained at low pH and these are not discussed in the manuscript. Overall, the data presented offers no mechanistic insight beyond speculation.

3. In regards to the absence of data to support mechanistic interpretations, there are several experiments that could be done to support some of the mechanistic hypothesis presented in the paper: for example, mutations at positions E570 and R557 that do not engage with SB-366791 would be predicted to disrupt activation by RTx but not inhibition by SB-366791. Conversely, mutations at R575 and E693 would be predicted to favor activation by vanilloid agonists and also potentially de-stabilize channel closure.

Reviewer #3 (Remarks to the Author):

This study from Neuberger et al. presents structures of the human TRPV1 channel in apo form as well as bound to the SB competitive antagonist. The perceived novelty of the study is high, as although there are many structures available for TRPV1, there are no structures to my knowledge of the human receptor, and sequence identities are not very close to 100% for known structures. Perhaps more importantly, they present the first structure with the competitive antagonist SB-366791, a novel analgesic. Structures of the apo channel were determined in circularized lipid nanodiscs with two different lipid compositions, yielding high resolution structural information and a more complete atomic model from the soy lipid nanodiscs. The structure of the SB antagonist complex was determined in GDN detergent, and is basically identical to the resting/apo structures. The team finds that the ligand binds competitively with vanilloid agonists, intuitively competing with them, and they offer a clear mechanism for inhibition (comment below). Patch clamp electrophysiology supports function of the channel (response to chemical agonist and heat), and supports interactions suggested by the structural data. Overall this is a very straightforward study that is clearly presented and is perceived as an important advance for the field. My specific comments are all minor but those related to structural statistics are very important to address. Assuming these comments can be addressed (should be easy), I support publication in the journal.

1. Fig. 1b shows a high degree of variability in temperature dependence of activation. Are these different cells- is a lack of reproducible temperature control the basis for apparent variation in the responses? Some start to activate at 30C, and some don't start to activate until 40C. The main text around lines 88-89 made me think the response would be highly reproducible. I do not suggest you repeat the experiment, just be more clear in the text about heterogeneity in results, if it is 'real.' Is this seen in other heating experiments with TRP channels? (I am more on the structure side).
2. Line 289, please list the ratio of PG:PC:PE.
3. From Table 1, and a more careful re-read of the Methods, the structure of the SB complex was determined in GDN, while apo TRPV1 structures were in nanodiscs with two different lipid compositions. Why did you switch to detergent for the SB complex? The difference in sample prep, even though it almost certainly has no consequence on your interpretations, should be mentioned in the main text, for the antagonist complex.
4. Lines 212 and 227, typo, should be 'pulling' and 'pulls'.
5. The mechanism of SB channel antagonism, beginning with lines 183-184. The details of the interaction and presentation in the figures are clear. However, the way the activity of the antagonist is presented is as if the antagonist is binding to activated channels and inducing them to close. Generally I (and others) think about an antagonist like SB instead as binding preferentially to a resting or other inhibited state- and stabilizing it- making activation less favorable. You can leave your description the way it is if you like, I just wanted to mention that the initial mechanistic idea was not intuitive to me (of binding to an activated state). Clearly, in a population experiment like whole cell patch, you can see it inhibit currents

(Fig 2a). I would still argue that the ligand is preferentially binding to the fraction of receptors that are in a resting state, and shifting the whole population in that direction over time.

Structural statistics:

1. Are the Ramachandran outliers justified?

2. Please check the PDB reports carefully, there are a bunch of little problems. I would be interested to see versions where the new ligand is included for that complex- probably needs full deposition to get this. These 3 reports are not clearly labeled as to which goes with which structure. One has bond angle problems in a couple of leucines, one has clashes with what I presume is a ligand ($\sim >1.3\text{\AA}$ overlap), POV and YBG have many bond length and bond angle outliers.

3. In case it is useful, we have had many problems with phenix elbow-generated CIF files. We have had much better luck using the GRADE server from Global Phasing (<https://grade.globalphasing.org/>) for CIF-file generation, starting with the PDB code if available or a smiles string from pubchem. This has worked for us for POV, for example.

We thank Reviewers for their excellent suggestions that have led to a significant improvement of this manuscript. We have made changes in the manuscript in accordance with the details outlined in our responses below.

Reviewer #1 (Remarks to the Author):

This article by Neuberger et al. reports three cryoEM structures of the human TRPV1 channel for the first time in apo and inhibitor-bound states. The structures reveal the SB-366791 binding pocket, which is validated by mutagenesis and electrophysiology. Based on these observations and structural analyses, a mechanism of SB-366791 inhibition is proposed. The experiments are carefully done and the manuscript is well written. Although several structures of TRPV1 homologs have already been published, this study provides very exciting new structural insights into the structural pharmacology of human TRPV1. Given the importance of TRPV1 in human biology and its emerging role in pain modulation, I recommend the publication of this work.

We thank Reviewer #1 for the generous assessment of our work.

I only have a few minor comments for the authors to consider adding into the discussion.

1. How similar is the apo state of human TRPV1 compared to that of other homologs in terms of RMSD?

We have made a supplementary figure that shows human, squirrel, and rat TRPV1 and their superposition (new Supplementary Figure 2). While they do look similar at the first glance, the RMSD values are relatively high (2.26 Å for human versus rat and 2.39 Å for human versus squirrel) because of substantial differences in the S1-S2, S2-S3 and S5-P loops as well as in the C-terminus. The corresponding information has been added to the text (page 6, last paragraph).

2. Line 166. What is the rationale to mutate L547 into Arg but not Ala (while mutating Y511 and L515 into Ala)?

While the manuscript was in review, we actually finished experiments with another mutant, T550A (now added to Figure 1). At the time we designed all our mutants, we were thinking about diversifying our scanning mutagenesis and L547 just happened to be the position where we decided to introduce a substitution different from alanine.

3. Is the proposed inhibition mechanism of SB-366791 different from that of capsazepine?

No, we think the mechanism is the same. We measured distances between R557 and E570 as well as R575 and E693 (E692 in rat TRPV1) in capsazepine- (PDB ID: 5IS0)¹, SB-366791- and RTX-bound (PDB ID: 7LQZ)² structures and they seem to be consistent with this idea. Indeed, the R557-E570 distance in the RTX-bound structure (< 4 Å) is relatively short and permissive for electrostatic interaction, while in SB-366791-bound (~6.1 Å) and capsazepine-bound (~4.6 Å) structures these distances are too large for strong interaction. Similarly, the R575-E693 (R575-E692) distances in SB-366791- and capsazepine-bound structures are shorter than 4 Å allowing electrostatic interaction, while in the RTX-bound structure the R575-E692 distance is > 12 Å and is too large to assume any meaningful interaction. The corresponding information has been added to the text (page 10).

4. It is definitely not necessary to re-collect a dataset, but is there a reason why the SB-366761-bound structure is determined in detergent, rather than in nanodisc?

Typically, the yield of the human TRPV1 preps is low and not sufficient for nanodisc reconstitution. When we aimed to solve apo state structures, the yield was exceptionally high, and we went ahead and did reconstitution in two different nanodiscs and solved the corresponding structures. When we aimed to solve a structure in complex with SB-366761, the yield was low, and we did not have enough protein to carry out reconstitution. Nevertheless, the amount of protein was sufficient for making grids in GDN detergent and we solved the corresponding structure in complex with SB-366761. After we realized that the resolution of the structure in GDN is actually higher than in nanodiscs, we decided not to pursue solving the SB-366761-bound structure in nanodiscs, given that the previous closed-state structures of rat TRPV1 did not show differences, except the resolution^{3,4}.

Chia-Hsueh Lee

Reviewer #2 (Remarks to the Author):

The TRPV1 is a cation-selective ion channel that is expressed in pain-sensing neurons and is activated by heat, vanilloids, protons, and by inflammatory conditions. TRPV1 inhibitors could potentially be used as non-addictive analgesics by stopping some forms of pain at the source – the nociceptive neurons.

Multiple cryo-EM structures of TRPV1 of a few animal species have been obtained in the presence of an antagonist and agonists that bind to the vanilloid pocket. In the present manuscript, a structure of the full-length human TRPV1 channel is obtained in the apo state and in the presence of the antagonist SB-366791. The structures are of high quality, displaying well-defined maps for multiple lipids that had been previously observed. There is also a clear non-protein density at the vanilloid pocket that can be assigned to SB-366791, consistent with previous studies showing that SB-366791 acts as a competitive inhibitor for activation by the vanilloid agonist capsaicin. Although the structural work is solid, and the data of high quality, the new insight provided by this manuscript is limited, and in my opinion would not warrant publication in its present form. My specific concerns are as follows:

We thank Reviewer #2 for the generous comment on the high quality of our data but respectfully disagree with the assessment of novelty. We present the first structures of human TRPV1, which appear to have similar architecture to the previously published structures of rat and squirrel TRPV1. Nevertheless, the structure of human TRPV1 reveals substantial and important differences (see response to comment 1 of Reviewer #1), which are likely critical for the ion channel function and will certainly need to be considered during structure-guided drug design.

1. I recognize the potential value of having a structure of the human TRPV1 channel, as opposed to structures of rodent orthologues. However, the human and rat TRPV1 channels are almost identical (83% sequence identity). Importantly, the regions that contain the highest number of amino acid sequence differences between human and rodents seem to be located at regions that were not well defined in the structures in this manuscript. No discussion is provided to address whether there are indeed any significant differences between the human structures and those of animals, and if any of these differences could be of any mechanistic or biological relevance. If no noteworthy differences can be found in the regions that were well-defined in maps, I would argue that the advancement represented by these human structures is rather incremental.

It can equally well be argued that 83% sequence identity is the evidence for the orthologs to be rather very different. In ion channels, a single substitution can cause dramatic changes in function. For example, the RNA editing that results in a single residue (R to Q) substitution in GluA2 AMPA receptor subunit converts the susceptible to polyamine block and Ca^{2+} -permeable channels to polyamine block-resistant and Ca^{2+} -impermeable, with enormous consequences for neurophysiology. The difference in sequence between the edited and non-edited versions of GluA2 (1 residue) is only ~0.1% for the 883-residue protein. We calculated the difference between the amino acid sequences of human, rat, and squirrel TRPV1 for the regions resolved in the available structures and it is about 10% (90% sequence identity, see alignment in Supplementary Fig. 5). While this is indeed less than 17% for the full-length sequences, it is two orders of magnitude more than ~0.1% that leads to dramatic functional consequences in GluA2. An example for TRPV1 protein is a conversion of weakly temperature-sensitive squirrel TRPV1 into highly temperature-sensitive rat TRPV1-like channel with only one mutation, N126S, in the structurally resolved region of ankyrin repeats^{2,5}. Furthermore, the K156A mutation in human TRPV1 was shown to completely eliminate the channel activity in the presence of high concentrations (2 mM) of ATP⁶. These two last examples have now been mentioned in the text (page 6, last line) as evidence of significant differences between the orthologs, also supported by the relatively high values of the RMSD mentioned in response to the first comment of Reviewer #1.

2. There was already evidence pointing to SB-366791 binding to the vanilloid pocket, and also showing that it also antagonizes channel activation by acidic pH. The mutagenesis results in Figure 2 are therefore unsurprising, especially considering that those same mutations also affect activation by vanilloids as shown in this and other studies. The specific binding pose identified in the experimental map in this study could be valuable. However, the conformation of the channel bound to SB-366791 appears to be indistinguishable from the apo state of the human and rodent structures, as well as the structure bound to the inhibitor capsazepine that also targets the vanilloid site – no discussion is provided to indicate the extent to which these structures are similar to each other, but from the cartoon in Figure 5d, the same interactions are highlighted in apo and SB-366791-bound states. Therefore, the specific mechanism of inhibition of SB-366791 remains in question: whereas it could act competitively to displace vanilloid agonists, it is less clear how it would inhibit channel activation by protons and other non-vanilloid activators. Structures of TRPV1 have been obtained at low pH and these are not discussed in the manuscript. Overall, the data presented offers no mechanistic insight beyond speculation.

Again, the relatively high RMSD values for the overall superpositions of structures of different orthologs (see the previous point and the response to the first comment of Reviewer #1) suggest substantial differences that may affect the molecular mechanisms of TRPV1 regulation. Even if the mechanisms are the same in principle, small molecular differences might cause tuning of kinetic parameters that are extremely important for physiological functions. Whether activation by protons and non-vanilloid activators is different in human TRPV1 versus TRPV1 in other species is definitely an important and interesting question. At the moment, we are not aware of any published structures, which would be open by low pH without help of agonists (despite lowering pH in physiological conditions can open the channel easily). Correspondingly, discussion of the structural basis of TRPV1 pH gating in the context of our study appears to be premature. Hopefully, this question will be addressed soon by solving open-state structures of human, rat, or squirrel TRPV1 at low pH but currently it is beyond the scope of our study.

3. In regards to the absence of data to support mechanistic interpretations, there are several experiments that could be done to support some of the mechanistic hypothesis presented in the paper: for example, mutations at positions E570 and R557 that do not engage with SB-366791 would be predicted to disrupt activation by RTx but not inhibition by SB-366791. Conversely, mutations at R575 and E693 would be predicted to favor activation by vanilloid agonists and also potentially de-stabilize channel closure.

We are grateful to Reviewer #2 for the excellent suggestion. Mutagenesis in the regions involved in channel gating is often very tricky because mutations can affect gating instead of inhibition only. Nevertheless, we made alanine substitution of all four suggested residues. One substitution, R575A, rendered TRPV1 non-functional and we have not been able to record currents from this mutant. Another two substitutions, R557A and E570A, did produce currents but only in response to acidic pH. Capsaicin did not induce any currents suggesting that the vanilloid site is likely destroyed in these mutants. In agreement with this assessment, SB-366791 did not induce any inhibition of current induced by changes in pH (see Figure below).

Figure. a, Exemplar current-voltage plots in response to a voltage ramp before (baseline) and after application of 10 μM capsaicin to HEK293 cells expressing wild-type hTRPV1 and mutants (R557A and E570A). **b**, Quantification of peak current for wild-type and mutant hTRPV1 evoked by 10 μM of capsaicin (Cap), normalized to baseline. Data are shown as mean \pm SEM. Points represent recordings from individual cells. (WT, $n = 6$; R557A, $n = 7$; E570A, $n = 7$). Statistical analysis: Ordinary one-way ANOVA with Dunnett's multiple comparisons test. **** $P < 0.0001$. **c**, Exemplar whole-cell currents recorded from hTRPV1-expressing HEK 293 cell in response to a voltage ramp at pH5 in the presence of different concentrations of SB-366791. **d**, Concentration-dependence of wild-type and mutant hTRPV1 inhibition by SB-366791 at pH 5, measured at +80 mV. Data are mean \pm SEM. Lines represent fits to the Hill equation (WT, $n = 7$) or connect data points (R557A, $n = 6$; E570A, $n = 6$).

In contrast to the three non-informative mutants, E693A had both agonist activation and SB-366791 inhibition preserved. In excellent agreement with the Reviewer's prediction, activation by capsaicin remained virtually unchanged ($EC_{50} = 0.207 \mu\text{M}$ for E693A compared to $EC_{50} = 0.234 \mu\text{M}$ for WT), while inhibition by SB-366791 was significantly weakened by the mutation ($IC_{50} = 0.100 \mu\text{M}$ for E693A compared to $IC_{50} = 0.021 \mu\text{M}$ for WT), strongly supporting our putative structural mechanism of inhibition proposed in Figure 5. However, we fully acknowledge that our proposed molecular mechanism of hTRPV1 allosteric modulation by small molecules needs to be further experimentally explored and validated in future studies. We have therefore rephrased the wording in the mechanism section to further emphasize the hypothetical nature of our mechanistic interpretation.

Supplementary Figure 6. a, Exemplar whole-cell current traces evoked by a voltage ramp in HEK 293 cells expressing hTRPV1-wild type and E693A mutant in the presence of capsaicin from 0.01 to 10 μM . **b**, Concentration-dependence of hTRPV1 wild-type and E693A mutant activation by capsaicin measured at +80 mV. Data are shown as mean \pm SEM. Lines represent fits to the Hill equation (WT, $n = 7$; E693A, $n = 6$). **c**, Quantification of half-maximal effective capsaicin concentration (EC_{50}) for hTRPV1-wild type and E693A mutant. Data are shown as mean \pm SEM. Points represent EC_{50} estimates from Hill fits of recordings from individual cells. (WT, $n = 7$; E693A, $n = 6$). Statistical analysis: t -test. $P=0.7099$. **d**, Exemplar whole-cell currents recorded from hTRPV1-wild type and E693A mutant in HEK 293 cells at pH5 in response to a voltage ramp in the presence of different concentrations of SB-366791. **e**, Concentration-dependence of wild-type and E693A mutant hTRPV1 inhibition by SB-366791 measured at +80 mV. Data are mean \pm SEM. Lines represent fits to the Hill equation (WT, $n = 7$; E693A, $n = 6$). **f**, Quantification of half-maximal inhibitory concentration (IC_{50}) of SB-366791 for hTRPV1-wild type and E693A mutant. Data are shown as mean \pm SEM. Points represent IC_{50} estimates from Hill fits of recordings from individual cells. (WT, $n = 7$; E693A, $n = 6$). Statistical analysis: t -test.

$P < 0.0001$. **g**, Quantification of SB-366791-resistant current for hTRPV1-wild type and E693A mutant. Data are shown as mean \pm SEM. Points represent minimal current estimates from Hill fits of recordings from individual cells. (WT, $n = 7$; E693A, $n = 6$). Statistical analysis: t -test. $P = 0.0004$.

A new supplementary figure describing the E693A experiments (Supplementary Figure 6) as well as the corresponding text (page 11, last paragraph) have been added to the manuscript.

Reviewer #3 (Remarks to the Author):

This study from Neuberger et al. presents structures of the human TRPV1 channel in apo form as well as bound to the SB competitive antagonist. The perceived novelty of the study is high, as although there are many structures available for TRPV1, there are no structures to my knowledge of the human receptor, and sequence identities are not very close to 100% for known structures. Perhaps more importantly, they present the first structure with the competitive antagonist SB-366791, a novel analgesic. Structures of the apo channel were determined in circularized lipid nanodiscs with two different lipid compositions, yielding high resolution structural information and a more complete atomic model from the soy lipid nanodiscs. The structure of the SB antagonist complex was determined in GDN detergent, and is basically identical to the resting/apo structures. The team finds that the ligand binds competitively with vanilloid agonists, intuitively competing with them, and they offer a clear mechanism for inhibition (comment below). Patch clamp electrophysiology supports function of the channel (response to chemical agonist and heat), and supports interactions suggested by the structural data. Overall this is a very straightforward study that is clearly presented and is perceived as an important advance for the field. My specific comments are all minor but those related to structural statistics are very important to address. Assuming these comments can be addressed (should be easy), I support publication in the journal.

We thank Reviewer #3 for the generous assessment of our work.

1. Fig. 1b shows a high degree of variability in temperature dependence of activation. Are these different cells- is a lack of reproducible temperature control the basis for apparent variation in the responses? Some start to activate at 30C, and some don't start to activate until 40C. The main text around lines 88-89 made me think the response would be highly reproducible. I do not suggest you repeat the experiment, just be more clear in the text about heterogeneity in results, if it is 'real.' Is this seen in other heating experiments with TRP channels? (I am more on the structure side).

Indeed, the observed variability is typical for temperature-induced activation of human and other heat-sensitive TRPV1 orthologues in our hands. The variability is reflected in the text in the relatively high reported SEM value for \$Q_{10}\$ (\$22.5 \pm 10.2\$ ). While the exact reason for the observed variability is unclear, we think it is linked to the fact that hTRPV1 expresses very quickly. To contain this, and avoid potential toxicity associated with high basal current, we use only 100 ng of plasmid (the minimum amount we can use) and record 12-16 hours post transfection (data reported in M&M). This method allows us to record currents of reasonable amplitude and is the best recording outcome we could achieve with this channel.

2. Line 289, please list the ratio of PG:PC:PE.

The POPC:POPE:POPG ratio (3:1:1) has now been provided in the text of Methods section.

3. From Table 1, and a more careful re-read of the Methods, the structure of the SB complex was

determined in GDN, while apo TRPV1 structures were in nanodiscs with two different lipid compositions. Why did you switch to detergent for the SB complex? The difference in sample prep, even though it almost certainly has no consequence on your interpretations, should be mentioned in the main text, for the antagonist complex.

See also a response to the last comment of Reviewer #1. In essence, when we aimed to solve a structure in complex with SB-366761, the yield of protein prep was insufficient to carry out reconstitution but enough for making grids with protein still in GDN detergent. After we solved the corresponding structure in complex with SB-366761, we realized that the resolution is actually higher than that for structures in nanodiscs. As a result, we decided not to pursue solving the SB-366761-bound structure in nanodiscs, given that previous closed-state structures of rat TRPV1 did not show any differences, except in resolution^{3,4}. We have now explicitly mentioned GDN when introducing the structure in complex with SB-366761 (page 7, line 8 from the bottom).

4. Lines 212 and 227, typo, should be 'pulling' and 'pulls'.

Thank you for noticing. The typos have been fixed.

5. The mechanism of SB channel antagonism, beginning with lines 183-184. The details of the interaction and presentation in the figures are clear. However, the way the activity of the antagonist is presented is as if the antagonist is binding to activated channels and inducing them to close. Generally I (and others) think about an antagonist like SB instead as binding preferentially to a resting or other inhibited state- and stabilizing it- making activation less favorable. You can leave your description the way it is if you like, I just wanted to mention that the initial mechanistic idea was not intuitive to me (of binding to an activated state). Clearly, in a population experiment like whole cell patch, you can see it inhibit currents (Fig 2a). I would still argue that the ligand is preferentially binding to the fraction of receptors that are in a resting state, and shifting the whole population in that direction over time.

We view this mechanism as competition with whatever is bound at the vanilloid site. In the apo state, this site harbors PI, and SB-366761 applied in the absence of ligands will simply outcompete PI. In the agonist-activated state, the site harbors capsaicin, and SB-366761 applied in the presence of capsaicin will likely outcompete this agonist and bring the channel back to the closed state. However, we absolutely agree with Reviewer #3 that even in the presence of capsaicin, there is a fraction of TRPV1 that resides in the closed state, and SB-366761 has a chance to outcompete the ligand (PI or capsaicin) while the channel is closed. We have now mentioned this possibility in the text (page 12, second paragraph).

Structural statistics:

1. Are the Ramachandran outliers justified?

We think that they are justified, and their number (0.2%) is typical for structures at this resolution.

2. Please check the PDB reports carefully, there are a bunch of little problems. I would be interested to see versions where the new ligand is included for that complex- probably needs full deposition to get this. These 3 reports are not clearly labeled as to which goes with which structure. One has bond angle problems in a couple of leucines, one has clashes with what I presume is a ligand (~>1.3Å overlap), POV and YBG have many bond length and bond angle outliers.

As suggested in the next point, we used Global Phasing to generate new CIF files. This dramatically improved the ligand geometry and eliminated the corresponding clashes. The validation report names have now included structure identifiers.

3. In case it is useful, we have had many problems with phenix elbow-generated CIF files. We have had much better luck using the GRADE server from Global Phasing (<https://grade.globalphasing.org/>) for CIF-file generation, starting with the PDB code if available or a smiles string from PubChem. This has worked for us for POV, for example.

We are very thankful to Reviewer #3 for the great suggestion. We remade the CIF files using GRADE, which significantly improved the refinement statistics.

References

- 1 Gao, Y., Cao, E., Julius, D. & Cheng, Y. TRPV1 structures in nanodiscs reveal mechanisms of ligand and lipid action. *Nature* **534**, 347-351 (2016).
- 2 Nadezhdin, K. D. *et al.* Extracellular cap domain is an essential component of the TRPV1 gating mechanism. *Nature communications* **12**, 1-8 (2021).
- 3 Liao, M., Cao, E., Julius, D. & Cheng, Y. Structure of the TRPV1 ion channel determined by electron cryo-microscopy. *Nature* **504**, 107-112, doi:10.1038/nature12822 (2013).
- 4 Gao, Y., Cao, E., Julius, D. & Cheng, Y. TRPV1 structures in nanodiscs reveal mechanisms of ligand and lipid action. *Nature* **534**, 347-351, doi:10.1038/nature17964 (2016).
- 5 Laursen, W. J., Schneider, E. R., Merriman, D. K., Bagriantsev, S. N. & Gracheva, E. O. Low-cost functional plasticity of TRPV1 supports heat tolerance in squirrels and camels. *Proc Natl Acad Sci U S A* **113**, 11342-11347, doi:10.1073/pnas.1604269113 (2016).
- 6 Shimizu, T., Yanase, N., Fujii, T., Sakakibara, H. & Sakai, H. Regulation of TRPV1 channel activities by intracellular ATP in the absence of capsaicin. *Biochimica et Biophysica Acta (BBA)- Biomembranes* **1864**, 183782 (2022).

REVIEWERS' COMMENTS

Reviewer #1 (Remarks to the Author):

The authors have satisfactorily addressed my concerns and provided additional discussion in the revised manuscript. Therefore, I recommend that the manuscript be published.

Reviewer #2 (Remarks to the Author):

The authors have addressed most concerns and the manuscript is significantly improved. I consider that the points below should still be addressed before publication:

1) "Since these regions were reported to be involved in temperature and ligand gating of TRP channels(19,33,34), the observed conformational variability as well as individual substitutions in other parts of the protein may explain differences in thermo-sensitivity between different species ..."

References cited (19,33,34) lack any experimental data concerning temperature-dependence of TRPV1 channel activation. Also, to my knowledge, there is no experimental data linking the S1-S2 or S2-S3 helices to the mechanism of TRPV1 channel activation by heat. Those statements need to be removed or re-phrased for accuracy. Appropriate references are required concerning the role of the pore domain in channel activation by heat: doi: 10.1038/nn.2552; doi: 10.1073/pnas.1717192115.

2) "Furthermore, the K156A mutation in human TRPV1 was shown to completely eliminate channel activity in the presence of 2 mM of ATP (36)."

The relevance of the statement above in the context of the manuscript is unclear: are rat TRPV1 channels containing the same mutation still active in the presence of ATP? Clarification or removal of the reference is needed.

3) "Binding of SB-366791 to the vanilloid site suggests that it acts as a competitive antagonist."

The discussion in the manuscript regarding the mechanism of inhibition by SB-366791 does not satisfactorily acknowledge previous experimental work providing biochemical evidence that SB-366791 acts as a competitive inhibitor of capsaicin activation (reference #20 in manuscript). That same study also provides experimental evidence using patch-clamp electrophysiology that SB-366791 inhibits human TRPV1 channels stimulated by protons. The manuscript text should be adjusted to acknowledge previous work on the mechanism of inhibition by this same antagonist in human TRPV1.

Reviewer #3 (Remarks to the Author):

The authors have addressed all of my concerns and I support publication of the article.

We thank Reviewers #1 and #2 for their previous comments and Reviewer #2 for the new suggestions. To address new suggestions of Reviewer #2, we have made changes in the text outlined in our responses below.

Reviewer #1 (Remarks to the Author):

The authors have satisfactorily addressed my concerns and provided additional discussion in the revised manuscript. Therefore, I recommend that the manuscript be published.

Reviewer #2 (Remarks to the Author):

The authors have addressed most concerns and the manuscript is significantly improved. I consider that the points below should still be addressed before publication:

1) "Since these regions were reported to be involved in temperature and ligand gating of TRP channels(19,33,34), the observed conformational variability as well as individual substitutions in other parts of the protein may explain differences in thermo-sensitivity between different species ..."

References cited (19,33,34) lack any experimental data concerning temperature-dependence of TRPV1 channel activation. Also, to my knowledge, there is no experimental data linking the S1-S2 or S2-S3 helices to the mechanism of TRPV1 channel activation by heat. Those statements need to be removed or re-phrased for accuracy. Appropriate references are required concerning the role of the pore domain in channel activation by heat: doi: 10.1038/nn.2552; doi: 10.1073/pnas.1717192115.

We are thankful to Reviewer #2 for noticing this obvious oversight. As requested, we removed references 19, 33 and 34 and added the two references concerning the role of the pore domain in channel activation by heat as well as two more references (doi.org/10.1038/s41594-021-00616-3; doi.org/10.1038/s41467-022-30602-2). Since we refer to TRP channels overall and not TRPV1, we introduced additional references showing involvement of the mentioned regions in temperature gating (doi.org/10.1038/s41594-018-0108-7; doi: 10.1038/s41594-021-00615-4). Since we refer not only to temperature gating but also to ligand gating, we added references that show involvement of S1-S2 (doi: 10.1038/s41594-018-0108-7) and S2-S3 (doi:10.1038/s41467-018-04828-y; doi:10.7554/eLife.49572) loops in ligand gating (see also the following review doi: 10.3389/fphar.2022.900623).

2) "Furthermore, the K156A mutation in human TRPV1 was shown to completely eliminate channel activity in the presence of 2 mM of ATP (36)."

The relevance of the statement above in the context of the manuscript is unclear: are rat TRPV1 channels containing the same mutation still active in the presence of ATP? Clarification or removal of the reference is needed.

The entire sentence as well as the corresponding reference have been removed as suggested.

3) "Binding of SB-366791 to the vanilloid site suggests that it acts as a competitive antagonist."

The discussion in the manuscript regarding the mechanism of inhibition by SB-366791 does not

satisfactorily acknowledge previous experimental work providing biochemical evidence that SB-366791 acts as a competitive inhibitor of capsaicin activation (reference #20 in manuscript). That same study also provides experimental evidence using patch-clamp electrophysiology that SB-366791 inhibits human TRPV1 channels stimulated by protons. The manuscript text should be adjusted to acknowledge previous work on the mechanism of inhibition by this same antagonist in human TRPV1.

We are thankful to Reviewer #2 again for noticing this obvious oversight. We have now mentioned in the text that in full agreement with the mechanism of competitive inhibition, SB-366791 caused a rightward shift in the capsaicin concentration dependence of hTRPV1-mediated calcium uptake with no apparent change in the maximal response, which was also confirmed by Schild analysis (reference #20).

Reviewer #3 (Remarks to the Author):

The authors have addressed all of my concerns and I support publication of the article.